# Femtosecond time synchronization of optical clocks off of a flying quadcopter

Hugo Bergeron[1,2,3], Laura C. Sinclair [1,2,3], William C. Swann[1,2], Isaac Khader[1,2], Kevin C. Cossel[1,2], Michael Cermak[1,2], Jean-Daniel Deschênes[1,2] & Nathan R. Newbury[1,2]

Future optical clock networks will require free-space optical time-frequency transfer between flying clocks. However, simple one-way or standard two-way time transfer between flying clocks will completely break down because of the time-of-flight variations and Doppler shifts associated with the strongly time-varying link distances. Here, we demonstrate an advanced, frequency comb-based optical two-way time-frequency transfer (O-TWTFT) that can successfully synchronize the optical timescales at two sites connected via a time-varying turbulent air path. The link between the two sites is established using either a quadcopter-mounted retroreflector or a swept delay line at speeds up to $24\,\mathrm{ms}^{-1}$. Despite 50-ps breakdown in time-of-flight reciprocity, the sites' timescales are synchronized to $< 1\,\mathrm{fs}$ in time deviation. The corresponding sites' frequencies agree to $\sim 10^{-18}$ despite $10^{-7}$ Doppler shifts. This work demonstrates comb-based O-TWTFT can enable free-space optical networks between airborne or satellite-borne optical clocks for precision navigation, timing and probes of fundamental science.

[1] National Institute of Standards and Technology, 325 Broadway, Boulder, CO 80305, USA. [2] Université Laval, 2325 Rue de l'Université, Québec, QC G1V 0A6, Canada. [3] These authors contributed equally: Hugo Bergeron, Laura C. Sinclair. Correspondence and requests for materials should be addressed to L.C.S. (email: laura.sinclair@nist.gov) or to J.-D.D. (email: octosigconsulting@gmail.com) or to N.R.N. (email: nathan.newbury@nist.gov)

O ptical clock networks promise advances in precision navigation, time distribution, coherent sensing, relativity experiments, dark matter searches, and other areas[1–12]. Such networks will need to compare and synchronize clocks over free-space optical links between moving airborne or satellite-borne clocks. However, current comb-based optical two-way time–frequency transfer (O-TWTFT)[13–15] cannot support femtosecond clock synchronization in the presence of motion. Even modest closing velocities between clocks lead to many picoseconds of non-reciprocity in the two-way optical time-of-flight, and correspondingly large time synchronization errors. Here, we demonstrate an advanced comb-based O-TWTFT to synchronize clocks without penalty, despite strong effective closing velocities.

There are multiple challenges in implementing sub-femtosecond time–frequency distribution between moving clocks via free-space optical links. These challenges reflect and extend those faced by rf time–frequency transfer over free space[16–18] and optical time–frequency transfer via fiber optics[7–10,19–25]. First, because of turbulence and diffraction, the received free-space signals will be weak, vary strongly, and suffer frequent fades. These turbulence-induced effects are far less for rf links, because of the longer wavelength, or for fiber-optic links, because of the stable medium. Previous comb-based O-TWTFT has nevertheless overcome these turbulence effects to achieve femtosecond synchronization[13–15]. Here, we focus on the second critical challenge. Namely, the clock sites can move rapidly, leading to strong Doppler shifts and a complete breakdown in the reciprocity of the two-way time-of-flight. Consider even a terrestrial velocity of 30 m s$^{-1}$. The fractional Doppler shift of $10^{-7}$ must be suppressed by $10^{11}$ to synchronize clocks to $10^{-18}$ in frequency. At this same velocity, the non-reciprocal time-of-flight of 3 ps (due to the finite speed-of-flight) must be suppressed by $10^4$ to synchronize clocks to below 1 fs in time. This level of suppression is orders of magnitude beyond that achieved in rf time–frequency transfer. Moreover, it must be achieved, despite recurrent turbulence-induced signal fades.

Here, we demonstrate an advanced comb-based O-TWTFT that synchronizes clocks to within femtoseconds, despite motion. We synchronize two optical timescales connected via a quadcopter-mounted retroreflector or swept delay line over turbulent air paths at speeds up to 24 m s$^{-1}$. The synchronized clocks agree to $\sim 10^{-18}$ in frequency, despite $10^{-7}$ Doppler shifts, and to $< 1$ fs in time deviation, despite 50-ps breakdown in time-of-flight reciprocity.

## Results

**Advanced O-TWTFT System**. We synchronize two sites A and B each with a clock, or optical timescale, defined by the labeled pulses from a 200-MHz fiber frequency comb phase-locked to a $\sim 195$-THz local optical oscillator. (For a full atomic clock, this optical oscillator would be locked to an atomic transition.) Site A acts as the master site. Site B is synchronized to it by adjusting the phase of the site B frequency comb. Because a fully flyable optical clock/oscillator is currently unavailable, both sites are fixed and we instead change the distance between sites by bouncing the optical signals off a quadcopter-mounted retroflector or a rapidly swept delay line. In either case, the link also includes the 2- or 4-km free-space turbulent air path. As shown in Fig. 1, the link is folded to enable verification by a single short fiber link that directly connects the sites to provide out-of-loop verification of the time synchronization[13–15]. All O-TWTFT information traverses the 2–4-km open-path link as if the two clock sites were, in fact, separated by this distance.

The system uses a layered approach: TWTFT with a modulated communication channel for picosecond-level time transfer[26]

followed by TWTFT with coherent frequency comb pulses for femtosecond-level time transfer. Frequency-comb TWTFT uses linear-optical sampling (LOS) to achieve femtosecond uncertainty (with a 5-nanosecond ambiguity given by the $\sim 200$-MHz comb repetition rate). LOS requires the repetition rate of the two pulse trains transmitted across the link to differ by $\Delta f_r \sim 2$ kHz, which leads to inclusion of a third transfer comb X at Site A (see Fig. 1b.) The timing data from the communication channel and frequency-comb transfer are input into synchronization algorithms that, unlike ref. [13], resolve the 5-ns ambiguity on the comb pulse-by-pulse to generate four calculated timestamps $\{T_{AA}, T_{AB}, T_{BB}, T_{BA}\}$ (see the Methods section). These four calculated timestamps can be formally interpreted as in conventional two-way time-transfer wherein one signal departs site A at time $T_{AA}$, as recorded at site A, and arrives at site B at time $T_{AB} = T_{AA} + T_{A \to B} - \Delta t_{AB}$, as recorded at site B where $\Delta t_{AB}$ is the time offset between sites and $T_{A \to B}$ is the time-of-flight from A to B. A second signal departs site B at time $T_{BB}$ and arrives at site A at time $T_{BA} = T_{BB} + T_{B \to A} + \Delta t_{AB}$, where $T_{B \to A}$ is the time-of-flight from B to A. From these calculated timestamps, we find the clock time offset as

$$\Delta t_{AB} = \frac{1}{2}[T_{AA} - T_{AB} - T_{BB} + T_{BA}] + \frac{1}{2}[T_{A \to B} - T_{B \to A}] + \Delta T_{cal} \tag{1}$$

where $\Delta T_{cal}$ is an overall transceiver calibration.

To estimate the time offset at femtosecond levels, each of the three terms on the right-hand side of Eq. (1) must be similarly evaluated to femtosecond levels. The last term is the overall calibration offset between transceivers and is discussed in the Methods section. The middle term is the non-reciprocal (unequal) time-of-flight between signals traveling from site A to site B compared with signals traveling from site B to site A. With motion, this non-reciprocity is significant as discussed next. Finally, the first term is the linear sum of timestamps derived from the LOS of the arriving comb pulses. The closing velocity between the sites leads to strong Doppler shifts that, in turn, couple to system dispersion and can lead to significant systematic errors in these timestamps, if not corrected as described below.

**Velocity-dependent reciprocity breakdown**. For the case realized here experimentally, the two clocks are connected via a retro-reflector moving at closing velocity $V/2$ away from the clocks. The retroreflector is at a distance $L_A(t)$ from site A and $L_B(t)$ from site B. This scenario mimics time transfer via a moving, intermediate clock site—the solution presented here could be generalized to the alternate scenario of a stationary clock A and moving clock B with inclusion of the time dilation effect and choice of reference frame. Because of motion, the time-of-flight is non-reciprocal and given by[18]

$$T_{A \to B} - T_{B \to A} = (V/c)(T_{AB} - T_{BA} + \Delta t_{AB}) + (V/c^2)(L_A - L_B) \tag{2}$$

to first order in $V/c$, where $c$ is the speed of light. The first term arises because the link distance is sampled at different times (asynchronously) by the pulses traveling each direction since they do not necessarily depart their transceiver at the same time. The second term exists even for synchronous sampling and arises from the motion of the retroreflector. It can be derived from geometric considerations or more formally via Lorentz transformations. At a modest $V = 30$ m s$^{-1}$, 4-km link, and an asynchronous sampling of $T_{AB} - T_{BA} + \Delta t_{AB} \approx 0.5$ ms, the first term in (2) yields a non-reciprocal time-of-flight of 50 ps. At the same velocity and for a reflector located adjacent to one clock site $(L_A - L_B \sim 4$ km), as is the case with the swept delay line,

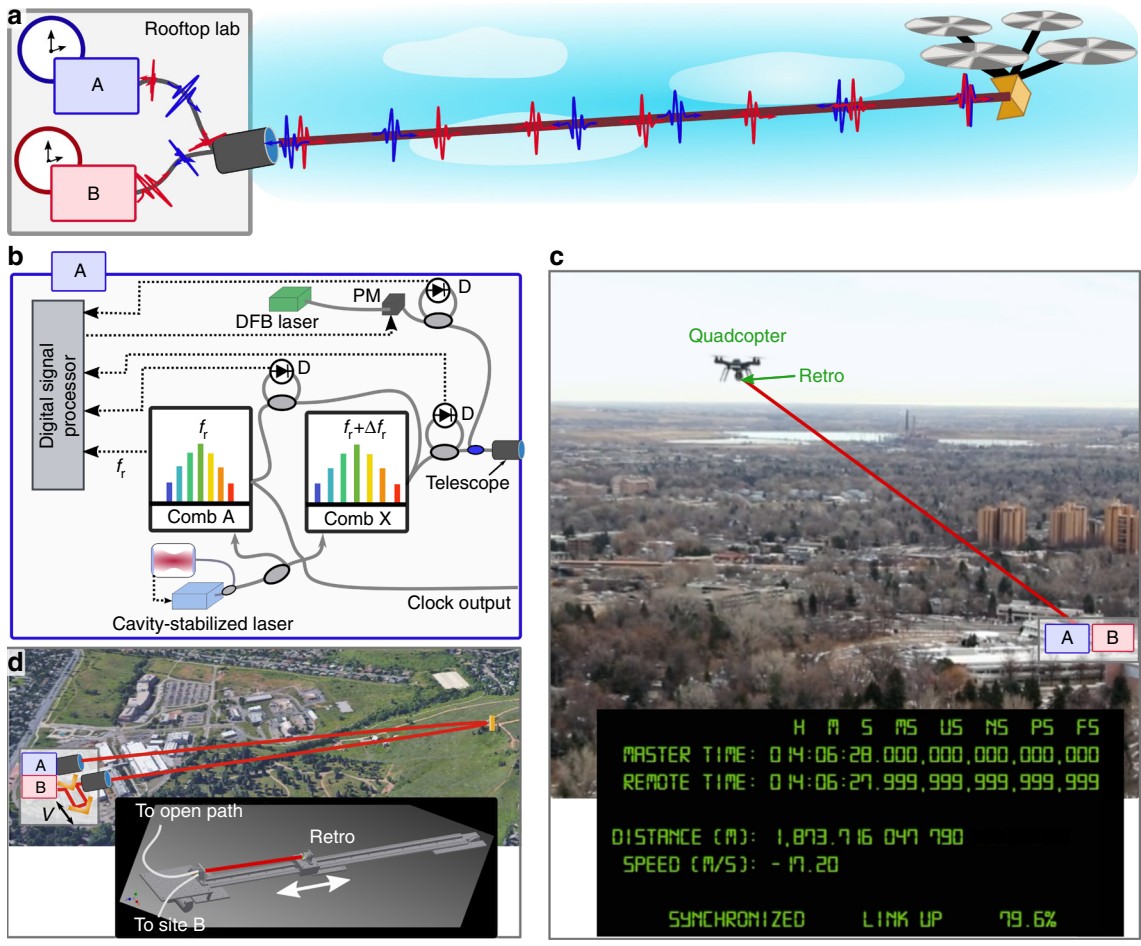

**Fig. 1** Experimental configuration for O-TWTFT demonstration via a moving platform. **a** Two optical timescales are synchronized over a folded link to a moving quadcopter-mounted retroreflector. The light is polarization multiplexed between site A and site B, co-located in a rooftop laboratory, and directed over the air to the quadcopter by a tracking telescope. **b** Schematic of master site A consisting of the comb A with repetition rate $f_r$, transfer comb X with repetition rate $f_r + \Delta f_r$, a cavity-stabilized laser, a phase-modulated distributed feedback (DFB) laser to support the optical communication channel and a digital signal processor. Remote site B is identical except that it contains only the remote comb B. In addition, the digital signal processor at site B applies feedback to remote comb B to synchronize the two sites. Gray lines: optical fiber; gray ovals: 50:50 couplers; blue oval: wavelength division multiplexer; D: balanced photodetector; dashed black lines: rf signals. **c** Images from Supplementary Movie 1. The real-time output includes the calculated times (from system turn-on), the round-trip propagation distance, the closing speed, and link status including the percentage of time without signal fades. **d** Additional experimental setup to synchronize the two sites over a 0–4 km free-space path to a fixed retroreflector and including an in-line 6-pass swept delay line that mimics high closing velocities (see inset)

the second term in (2) yields a non-reciprocal time-of-flight of 1.3 ps. Note that for a symmetrically located retroreflector, as is approximately true for the quadcopter, this term vanishes. We include these two non-reciprocity corrections in (1) to <100 as uncertainty by using the available O-TWTFT data to calculate the speed to 20 μm s$^{-1}$ uncertainty at 1-s averaging time. The speed is found from the rate-of-change of the measured time-of-flight (calculated via a different combination of timestamps) over three consecutive measurements. Errors due to acceleration are suppressed to <0.1 fs at even our maximum experimental acceleration of 7 g, both by the fast update rate (~2 kHz) of the O-TWTFT system and the use of centered derivatives to determine the speed at the correct time.

**Velocity-dependent systematic timing shifts**. The Doppler shifts of the received comb light are large ($10^{-7}$, or 20 MHz, at $V = 30$ m s$^{-1}$) and changing as $V$ is not constant. These Doppler shifts can couple with the system dispersion to cause distortions in the measured heterodyne signal between the incoming and local

comb light. These distortions can lead to picosecond-level timing errors in the calculated timestamps. To avoid this, we calculate the cross-ambiguity function[27] between the measured heterodyne signal and a frequency-shifted template of the expected zero velocity waveform. We find its peak in real time (< 300 μs) to <100 as uncertainty by use of a Fourier transform algorithm and the Nelder–Mead search algorithm[28].

**Synchronization feedback via an adaptive Kalman filter**. The final synchronization algorithms are implemented in a digital signal processing platform to generate an estimate of $\Delta t_{AB}$ in real time at a 2-kHz measurement rate. Under strong turbulence, signal fades block the exchange of comb pulses and the communication link, but these fades are usually of short duration, e.g., a few milliseconds. With motion, these signal fades can extend over longer durations because of the challenges of tracking the moving platform. Indeed, for the quadcopter data shown later, we suffer signal fades of many seconds. To synchronize despite these signal fades, we implement a Kalman-filter-based loop filter as in

ref. [29]. The Kalman filter provides optimal hold-over behavior during fades. It uses a two-element state vector, modeling, respectively, time error and frequency error. The state transition matrix is simply two integrators in order to accurately account for

the random frequency walk of the cavity-stabilized oscillators ($1/f^4$ phase noise). The measurement noise is modeled as white noise with a 5-femtosecond standard deviation at the 2-kHz update rate. The Kalman filter's output is sent to a standard

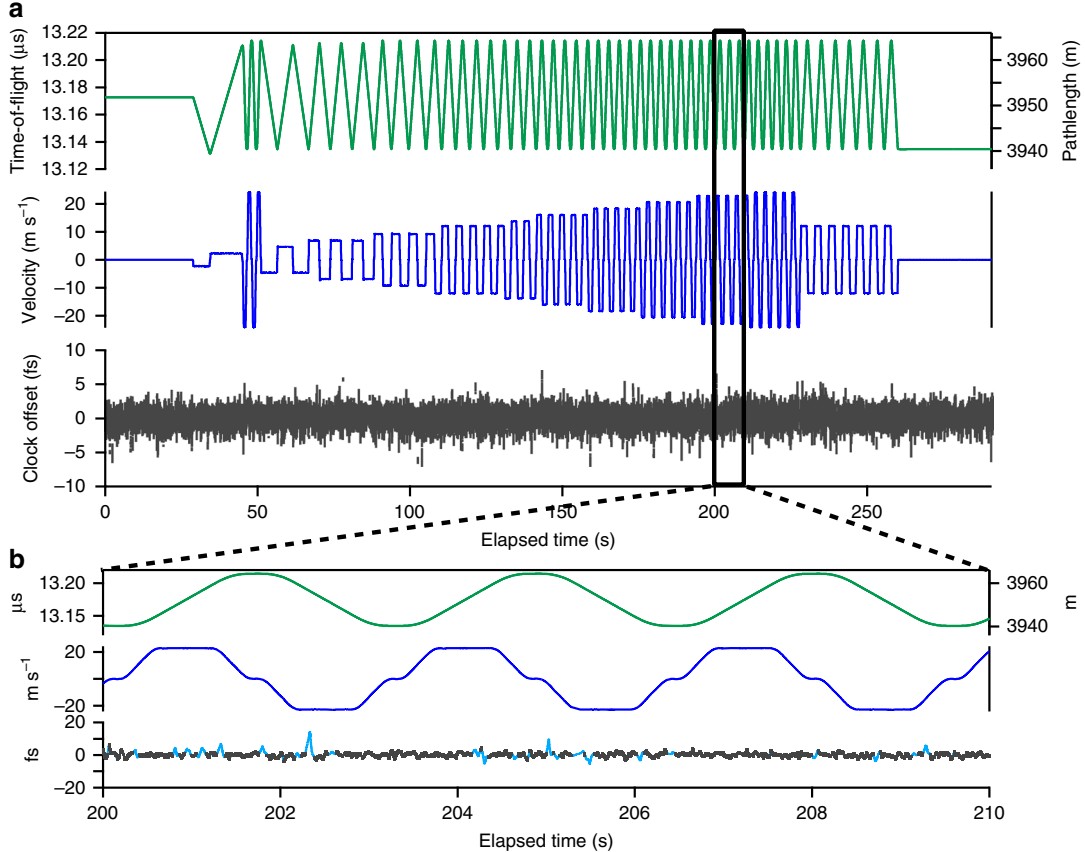

**Fig. 2** Synchronization over 4 km with the in-line swept delay line. **a** The time-of-flight and closing velocity are retrieved from the O-TWTFT data. The closing velocity varied from 0 m s$^{-1}$ to 24 m s$^{-1}$. The clock time offset is the out-of-loop verification. During active synchronization (i.e., no long fades) the standard deviation is 1.1 fs. All data are at the 2.2 kHz update rate. **b** Expanded view. The clocks' time offset is shown for all time (cyan) and only during active synchronization, i.e., no turbulence-induced fades (black line)

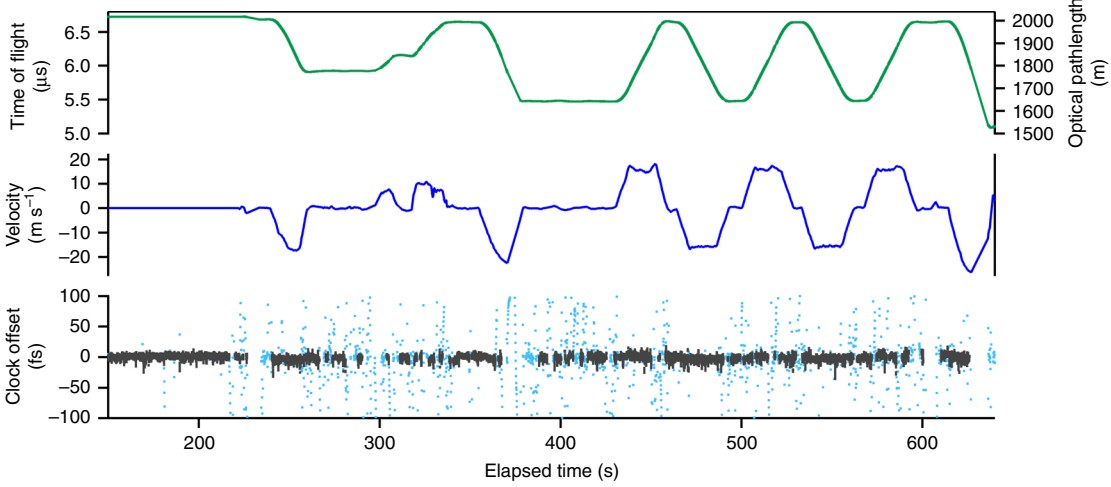

**Fig. 3** Synchronization results for a link to the flying quadcopter. The results shown are the time-of-flight (optical pathlength), closing velocity, and clocks' time offset as measured by the out-of-loop verification channel. The clock's time offset is given for periods of active synchronization (gray dots) at the full ~2 kHz update rate and for all times (cyan dots) at a 10 Hz sampling rate. The latter clearly shows the walkoff of the clocks during longer duration fades. The walkoff of the clock offset can exceed ~100 femtoseconds for fades longer than 1 s. The standard deviation is 3.7 fs for active synchronization at the ~2 kHz update rate (also see Supplementary Movie 1)

proportional-integrator controller which changes the Site B's frequency (via the offset between comb B and its local cavity-stabilized laser) to adjust the calculated time offset to zero. The combination of the measurement and clock noise levels sets the Kalman filter's effective bandwidth to 10–100 Hz, which limits the closed-loop bandwidth to the same value. During long fades, the clock time offset varies randomly according to the phase noise of the local cavity-stabilized laser. For our system phase noise, and in agreement with the Kalman filter model, the clock time offset can exceed 150 fs for fades longer than 1 s.

**Results of time synchronization with quadcopter and swept delay line.** Figure 2 shows time synchronization between the two sites A and B over a link that includes both 4 km of turbulent air and the swept delay line operated at closing velocities from 0 m s$^{-1}$ to ± 24 m s$^{-1}$. The time-of-flight, closing velocity, and calculated time offset are all returned from the O-TWTFT signals. In parallel, the clocks' time offset (i.e., arrival time of labeled optical comb pulses) is measured by the out-of-loop verification. When actively synchronized (outside of signal fades >20 ms), the clock times agree with a standard deviation of 1.1 fs at the full 2.2 kHz update rate. During brief signal fades due to atmospheric turbulence, the clocks' times walk off randomly as previously mentioned (cyan trace of Fig. 2b), but are resynchronized when the signal is reacquired.

Synchronization to a quadcopter-mounted retroreflector is shown in Supplementary Movie 1 and Fig. 3. The quadcopter provided a maximum 500-m optical pathlength change and a 20 m s$^{-1}$ (quadcopter-limited) maximum speed. Again, we see femtosecond-level synchronization with no evidence of speed-dependent bias. These data do show much longer fades due to the additional challenge of tracking the moving quadcopter, accomplished as in ref. [30].

**Analysis of time and frequency instability.** Figure 4 shows the time and modified Allan deviations. For these data, the swept delay line was operated for ~20 min at ± 24 m s$^{-1}$ with free-space links of 0, 2, and 4 km. The resulting time deviations, calculated from the out-of-loop verification, all remain below 1 fs for averaging times from 0.1 s (the inverse of the synchronization bandwidth) to 200 s and are essentially unchanged from a static 0-km shorted measurement. For the quadcopter, the time deviation remains at ~1 fs, elevated above the delay-line data due to the many long fades and calibration uncertainties associated with the tracking terminal. The relative fractional frequency instability (modified Allan deviation) for the swept-delay line data is below 10$^{-15}$ at a 1-s averaging and 10$^{-18}$ at 200-s averaging for all closing velocities. For the quadcopter data, it is 2 × 10$^{-15}$ at a 1-s averaging and 2 × 10$^{-17}$ at a 100-s averaging.

In conclusion, we demonstrate it is possible to synchronize two optical timescales to the femtosecond level despite closing velocities of up to 24 m s$^{-1}$. This level of synchronization is afforded by the calculation of all correction terms to sub-femtosecond level and by the similar suppression of velocity-dependent systematics due to the coupling of Doppler shifts and system dispersion. The synchronization implies syntonization as well, and thus the two sites' frequencies agree to ~10$^{-18}$ despite 10$^{-7}$ Doppler shifts.

This same approach should scale to the far greater closing velocities of future airborne and satellite-borne clocks, although this remains to be tested. Such high velocities will likely require implementation of IQ detection to permit continuous LOS of the comb pulses as the Doppler shifts exceed the comb repetition rate. In addition, ground-to-satellite links will suffer from non-reciprocity due to point-ahead effects, i.e., the separation of the

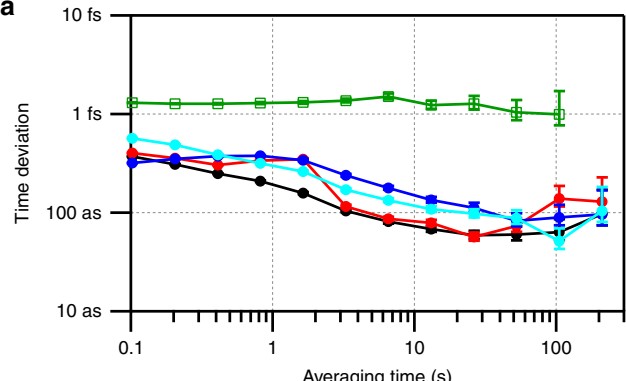

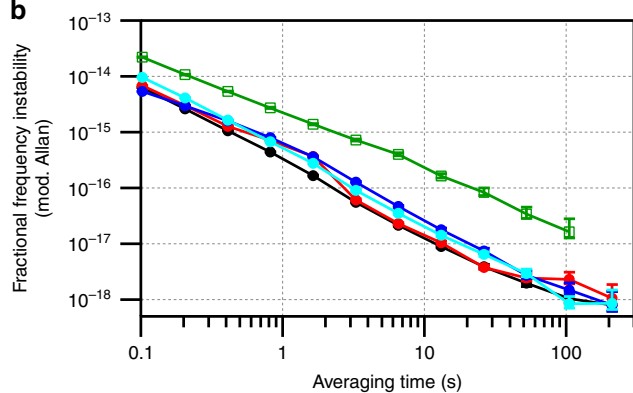

**Fig. 4** Time deviations and modified Allan deviations. **a** Time deviation for periods of active synchronization off the quadcopter with 0–20 m s$^{-1}$ motion (open green squares) and with the in-line swept delay line with ± 24 m s$^{-1}$ motion and a free-space pathlength of 0 m (red circles), 2 km (blue circles), and 4 km (cyan circles). Uncertainty bars are the statistical uncertainty on the calculated deviations. Also shown is the time deviation at 0 m s$^{-1}$ and 0 m free-space path (black circles). The elevated time deviation for the quadcopter data is due to the many signal fades and calibration uncertainties associated with the tracking terminal. The O-TWTFT synchronization bandwidth was 10 Hz. **b** Corresponding modified Allan deviation

up-link and down-link optical paths due to transverse motion, but theory and the ground-based experiments predict minimal impact[31–33].

The advanced comb-based O-TWTFT described here should support multiple applications that benefit from the ability to compare and synchronize high-performance clocks over free-space links to moving sites. One potential future application would be optically cross-linked synchronized clocks for global navigation satellite systems (GNSS). Such a configuration could reduce the number of full atomic clocks required for the constellation and enable higher performance navigation, although the latter would require other major advances. For scientific applications, comb-based O-TWTFT could enable free-space optical networks between airborne or satellite-borne platforms to support experiments ranging from relativity tests to dark matter detection.

## Methods

**Hardware**. The advanced comb-based O-TWTFT demonstrated here differs significantly in hardware, calibration, and algorithms from the previous comb-based O-TWTFT described in refs. [13–15]. The full extent of the hardware and algorithms is discussed in a follow-on article[34]. The overall transceiver structure is similar to ref. [13], but with major alterations to allow for the presence of motion. Briefly, the

overall dispersion in the transceivers was reduced 30-fold by use of dispersion compensation, the transceiver calibration was performed by a custom, integrated optical time domain reflectometer (OTDR), the rf system was redesigned to reduce multipath reflections due to impedance mismatches, the rf group delay were measured and digitally canceled prior to interferogram detection, and the digital signal processing hardware was redesigned to support the more extensive, real-time synchronization algorithms. A more detailed description of the redesigned transceiver is given in ref. [34].

**Synchronization algorithm**. The presence of motion in the O-TWTFT link required entirely new synchronization algorithms and implementation. The derivation of the four effective timestamps is lengthy and provided in ref. [34]. We briefly outline a different derivation here that leads to a single master synchronization equation, but does not provide the same physical insight as the use of the virtual calculated timestamps of Eq. (1).

At site A, the comb pulses arrive at the local defined reference plane at times $t_A = n_A f_r^{-1} + \tau_A$, where the integer $n_A$ labels the pulses, $f_r$ is the nominal defined repetition frequency and $\tau_A$ is the overall time offset that includes any integrated frequency error in the local comb. At site B, the comb B pulses arrive at the local defined reference plane at $t_B = n_B f_r^{-1} + \tau_B$, where again $\tau_B$ is the slowly varying clock offset. Note $\tau_A$ and $\tau_B$ are both here defined with respect to some common arbitrary ad hoc timescale; in the end, we are only concerned with their differences. For verification purposes, we locate the reference plane for both sites at the end of the optical fiber that is used for the out-of-loop time synchronization measurement so that this out-of-loop measurement directly yields $\Delta t_{AB} \equiv t_A - t_B$.

The linear-optical sampling (LOS) detection used in comb-based O-TWTFT requires the pulse trains of the two combs transmitted across the link have repetition rates differing by $\Delta f_r \sim 2$ kHz[13]. Therefore, we introduce a third, transfer comb X at the master site with repetition rate $f_r + \Delta f_r$ and time offset $\tau_x$, also phase-locked to the master optical oscillator.

We measure three heterodyne signals between the master, transfer, and remote combs, each consisting of a series of consecutive interferograms, i.e., short heterodyne pulse envelopes, as the pulses cross each other at a rate $\Delta f_r$. If we label each interferogram by an integer p with appropriate subscripts, the three signals are:

$$\tilde{I}_{AX}(t) \approx \sum_{p_{AX}} I_{AX}\left(t - t_{p_{AX}}\right) : \text{Master} \rightarrow \text{Transfer} ,$$

$$\tilde{I}_{BX}(t) \approx \sum_{p_{BX}} I_{BX}^{V}\left(t - t_{p_{BX}}\right) : \text{Remote} \rightarrow \text{Transfer}$$

$$\tilde{I}_{XB}(t) \approx \sum_{p_{XB}} I_{XB}^{V}\left(t - t_{p_{XB}}\right) : \text{Transfer} \rightarrow \text{Remote}$$

where $I_{BX}^{V}(t)$, $I_{XB}^{V}(t)$, and $I_{AX}(t)$ are the interferogram pulse shapes. The times $t_{p_{AX}}$, $t_{p_{BX}}$, and $t_{p_{XB}}$ are the arrival time of the $p_{AX}$th, $p_{BX}$th, and $p_{XB}$th interferograms. The time $t$ is some common ad hoc arbitrary timescale that is mathematically convenient, but drops out of the final synchronization equation. The first interferogram, $I_{AX}$, is generated from the local heterodyne mixing of the master and transfer comb at the master site. The middle interferogram, $I_{BX}^{V}$, is the heterodyne signal between the transmitted remote comb pulses and the transfer comb at the master site. The third interferogram, $I_{XB}^{V}$, is the heterodyne signal between the transmitted transfer comb and the remote comb at the remote site. We are ultimately interested in the times $t_{p_{AX}}$, $t_{p_{BX}}$, and $t_{p_{XB}}$ that define the centers of the successive interferograms since we will combine these timing data with the communication-based two-way time–frequency transfer to evaluate $\Delta t_{AB}$.

The interferograms' waveform depends on the Doppler shift of the incoming light as it couples to the relative chirp between the comb pulses thereby distorting the interferogram shape and causing systematic timing shifts. As discussed in the article, to avoid this, we use the ambiguity function, which suppresses this systematic to below 100 as.

Assuming successful suppression of this velocity-induced bias and following similar analysis as in ref. [13], the three times are

$$t_{p_{AX}} = \Delta f_r^{-1}\left\{p_{AX} - [f_r + \Delta f_r]\tau_X + f_r\tau_A\right\}$$

$$t_{p_{BX}} = \Delta f_r^{-1}\left\{p_{BX} - f_r T_{B\rightarrow A}\left(t_{p_{BX}}\right) + f_r\tau_B - [f_r + \Delta f_r]\tau_X\right\}$$

$$t_{p_{XB}} = \Delta f_r^{-1}\left\{p_{XB} + [f_r + \Delta f_r]T_{A\rightarrow B}\left(t_{p_{XB}}\right) + f_r\tau_B - [f_r + \Delta f_r]\tau_X\right\}$$

(3)

as measured with respect to the timescale $t$. In the system, however, the timestamps are instead measured against the local timescale at site A or B. Therefore, we use the relationships $t_{p_{AX}} = f_r^{-1}k_{p_{AX}} - \tau_A$, $t_{p_{BX}} = f_r^{-1}k_{p_{BX}} - \tau_A$, and $t_{p_{XB}} = f_r^{-1}k_{p_{XB}} - \tau_B$, where $\tau_{A(B)}$ are the time offsets of the site A(B) timescale from the ad-hoc timescale $t$ and $f_r^{-1}k$ is the definition of a local time. $k$ is the ADC sample number of the retrieved peak of the ambiguity function from the Nelder–Mead search, which is found with sub-sample precision. As a consequence, $k$ is not restricted to an integer ADC sample number. Note that the function $T_{A\rightarrow B}(t)$ is the time-of-flight for a signal that arrives at B at time $t$. To solve these equations for the time offset between sites, $\Delta t_{AB} = t_A - t_B$, we need the integer values, $p_{AX}$, $p_{BX}$, and $p_{XB}$, as extracted from the communication-based O-TWTFT. We also need the time-of-flight non-reciprocity, Eq. (2), which is briefly derived here.

From (3) and the substitution mentioned afterward, it is clear we are interested in the asymmetry $T_{A\rightarrow B}\left(f_r^{-1}k_{p_{XB}} - \tau_B\right) - T_{B\rightarrow A}\left(f_r^{-1}k_{p_{BX}} - \tau_A\right)$. We first must extrapolate the time-of-flight to a common measurement time. To this end and to first order in $V/c$, we write

$$T_{A\rightarrow B}\left(f_r^{-1}k_{p_{XB}} - \tau_B\right) = T_{A\rightarrow B}\left(f_r^{-1}k_{p_{BX}} - \tau_A\right) + (V/c)\left[f_r^{-1}k_{p_{XB}} - f_r^{-1}k_{p_{BX}} + \Delta t_{AB}\right]$$

(4)

where the second term is the asynchronous sampling contribution. Because of the finite speed of light, $T_{A\rightarrow B}(t) - T_{B\rightarrow A}(t) = (V/c)[L_A(t) - L_B(t)] + O((V/c)^2)$. Combined with (4), this yields the breakdown in reciprocity,

$$T_{A\rightarrow B}\left(f_r^{-1}k_{p_{XB}} - \tau_B\right) - T_{B\rightarrow A}\left(f_r^{-1}k_{p_{BX}} - \tau_A\right) = (V/c)\left[f_r^{-1}k_{p_{XB}} - f_r^{-1}k_{p_{BX}} + \Delta t_{AB}\right] \\ + (V/c^2)[L_A - L_B]$$

(5)

With this information, we can solve to find the time offset,

$$\Delta t_{AB} = \frac{1}{2 - V/c + \Delta f_r/f_r}\left\{\frac{\Delta f_r}{f_r^2}\left\{2k_{p_{AX}} - k_{p_{XB}} - k_{p_{BX}}\right\} + \frac{\Delta f_r}{f_r}T_{A\rightarrow B}\left(t_{p_{XB}}\right) \right. \\ + f_r^{-1}\left[p_{XB} + p_{BX} - 2p_{AX}\right] + 2\Delta T_{cal} \\ \left. + \frac{V}{c}\left(f_r^{-1}k_{p_{XB}} - f_r^{-1}k_{p_{BX}} + c^{-1}[L_A - L_B] + 2\Delta T_{cal}^{V}\right)\right\}$$

(6)

where we introduce two calibration terms, $\Delta T_{cal}$, $\Delta T_{cal}^{V}$, discussed below, and drop terms of order $(V/c)^2$ and higher throughout. In the real-time computation, the closing velocity, $V$, is found by a combination of centered numerical derivatives using the previous three measurements of $T_{A\rightarrow B}$ and $T_{B\rightarrow A}$, which assumes constant acceleration over $3/\Delta f_r \sim 1.5$ ms. A rough estimate of the time offset error due to a changing acceleration, i.e., jerk, is $\sim (1/6)(\dot{a}/c)(2\Delta f_r)^{-3}$. A 1 fs timing error is reached only at a jerk of $\dot{a} \sim 10^4$ g s$^{-1}$.

**Transceiver calibration**. The calibration term, $\Delta T_{cal}$, nominally reflects a time delay in the transceiver between the reference plane and the incoming pulse detection. However, in reality, each transceiver consists of multiple optical and rf paths between, for example, the optical oscillator, the frequency combs, the optical detection of the arriving frequency comb pulses, the various analog-to-digital converters, and throughout the communication-based O-TWTFT. Without motion, all these paths can be lumped into a single overall time delay. With motion and the resulting Doppler shifts, some delay paths must be corrected for velocity. As a result, the overall transceiver must be calibrated via a built-in rf-domain optical time domain reflectometer (OTDR) that measures the various required delays[34]. In a simplified view, the net result is that the calibration becomes velocity-dependent as $\Delta T_{cal} \rightarrow \Delta T_{cal} + (V/c)\Delta T_{cal}^{V}$.

## Data availability
The data that support the findings of this paper are available from the corresponding author upon reasonable request.

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

## Acknowledgements

This work was funded by the National Institute of Standards and Technology (NIST) and the Defense Advanced Research Projects Agency (DARPA) PULSE program. We thank Prem Kumar, Martha Bodine, Jennifer Ellis, and Kyle Beloy for helpful discussions.

## Author contributions

H.B., L.C.S., J.D.D., and N.R.N. determined the effects of motion and designed the algorithms. H.B. and J.D.D. implemented the digital signal processing. H.B., J.D.D., I.K., L.C.S., and W.C.S. acquired and analyzed the data. K.C.C. designed and implemented the tracking terminal necessary for quadcopter operation. W.C.S. designed and implemented the swept delay line. M.C. and K.C.C. assisted with hardware implementation and quadcopter operation. L.C.S., J.D.D., and N.R.N. prepared the paper.

## Additional information

**Competing interests:** The authors declare no competing interests.

