## [Peer Review File · Nature Communications]

Reviewers' comments:

Reviewer #1 (Remarks to the Author):

In the present manuscript the authors demonstrate a free space, optical two-way time and frequency transfer system that is robust against large change rates of the optical path length and achieves a precision compatible with optical atomic clocks.

The results presented in this work are certainly impressive and relevant to the time and frequency community. The claim that future optical clocks will require free-space optical time transfer may be bold but is not implausible. The free-space synchronisation and syntonisation technique developed by the authors and now demonstrated with a moving target is unique and world-leading.

The achievements presented in this paper are the cumulative result of several technological advancements, all of which have been (refs 13-15, 27, 28), or shortly will be (ref 32), published separately in greater detail. The challenges introduced by a moving clock site, which the authors choose to focus on, are not new, and the solution presented is conceptually straight-forward (see e.g. suggested reference below), albeit not necessarily easy to implement at this level of precision as the authors rightly point out.

As a whole, the achievement of femtosecond-level synchronisation via a flying quadcopter is a significant and noteworthy step towards practical free space, possibly satellite-aided, time and frequency transfer network for optical clocks.

The manuscript is generally well written and logical, though not always easy to understand. I have a number of specific technical comments – some of them may simply boil down to typing errors:

Line 51 – It is perhaps debatable whether the effects accounted for in this work should be called relativistic. Certainly, additional complications caused by clocks actually moving with respect to each other are not covered.

Line 106 – The authors may want to cite Jeremy Warriner, John Hein, and Tom Celano and Richard Beckman, "Real-Time Two-Way Time Transfer To Aircraft", 38th Annual PTTI Meeting 2006 (<https://tycho.usno.navy.mil/ptti/2006papers/paper39.pdf>) or another suitable reference.

Line 107 – It is somewhat misleading to refer only to the second term as being caused by the finite speed of light, as the first term clearly also scales with $1/c$.

Line 111 (also 46) - The number for the second term of the non-reciprocal correction, 1.3 ps, seems based on $L_A - L_B = 4$ km. Should this not be the *difference* of distances to the retroreflector, i.e. the presumably much shorter path segments between sites A and B, respectively, and the polarisation multiplexer? My understanding is that (most of) the free space path from each site to the mirror is shared.

Figure 1 – It would help with understanding, e.g. with the definition of events and timing calculations, if there was a schematic diagram of the full setup (sites A and B), not just site A.

Line 153 – Was the same tracking technology as in ref 28 used in this work? I assume so, but it might be worth stating explicitly.

Line 222 – I don't know the term "oracle timescale". I consulted two colleagues who didn't recognise it either. Is it an ad-hoc, arbitrary time scale whose relation to well-defined timescale is determined retrospectively?

Line 239 (equation 3) – Are the subscripted quantities on the left-hand side identical to those as in line 221 ($t_{AX}[p_{AX}]$ etc)?

Line 251 – Should the first time-of-flight on the right-hand side be labelled $T_{\{B \rightarrow A\}}$ rather than $T_{\{A \rightarrow B\}}$?

As a general remark, the authors do not refer to the accuracy of their technique, and they prefer to use the somewhat fuzzier term “precision” instead of “stability”. I wonder whether this could be clarified throughout the text, to bring it more in line with accepted metrological terminology. Compare for example Lisdat, C. et al. A clock network for geodesy and fundamental science. Nat. Commun. 7: 12443 doi: 10.1038/ncomms12443 (2016), which does use “precision” in places but also clearly analyses “stability” and “accuracy” separately.

Reviewer #2 (Remarks to the Author):

The paper by Bergeron et al extends their free-space synchronization technology to cover a dynamic, small target with fs-level precision. The authors have pioneered the free-space time-frequency transfer and synchronization field in the last few years, and the current work overcomes the broken reciprocity due to a moving target. What is new in this paper is that synchronization algorithm is modified to suppress time-of-flight reciprocity breakdown, which is unique challenges for aerial dynamic targets. Data supports well all involved methodologies such as drone tracking, linear optical sampling, and suppressed timing asymmetry. Modified Allan instability for moving quadcopter after synchronization could achieve 10-18 instability. I recommend to publish this paper with minor revision, as the impact of the result could benefit overall metrology and applications of frequency combs involving aerial moving targets.

Some minor suggestions:

1. I recommend authors to plot timing offset of clock during signal fading in Fig. 3 (as shown in Fig. 2).
2. As signal fades much frequently with quadcopter than with swept delay line, the role of Kalman filter based timing estimator is more important and more detailed discussion will be helpful.

Reviewer #3 (Remarks to the Author):

This paper describes new work in synchronizing two optical frequency references over free space in the presence of an effectively large relative velocity, and associated Doppler shift. This group is famous for pioneering work in the area of free-space optical time and frequency transfer, but previous work has focused on comparing stationary sites. The extension to moving sites is a natural one and novel in the literature warranting publication.

The main result of the paper is overcoming the large Doppler shift between two sites and still maintaining 1e-18 precision in the comparison. Instead of moving the sites themselves, which would add unnecessary additional complexity, they use a moving transponder in the form of a quadcopter. Since the main idea is to introduce a real Doppler shift, which the moving transponder will do, the approach appears to be valid and is used over a wide range of velocities, demonstrating its robustness. Results are consistent over the range tested, providing confidence. The authors also use a swept delay line as an initial proof of principle and get similar results. The paper is only interested in the time/frequency transfer aspect of the problem. Having operating optical frequency references that

are in motion is outside its scope.

The results appear to be valid and will be of interest to those working in this area. The work also appears to be the first of its kind, except for preparatory work by the same group.

The paper uses the Allan deviation to analyze the results and estimate final statistical uncertainty. These are the gold standards for this field and can't be argued with. The quadcopter data analysis appears to include the expected signal fades and drop outs caused by the intervening atmosphere and so addressing a more "real world" type of situation. As a result of the reasonable methodology and this statistical analysis, the conclusions appear to be valid.

I have the following comments that the authors should probably consider:

In general I found the paper easy to read except for the Method section. It seemed a little unusual to have it at the end, especially since it contains the key advancement of the paper, which is the mechanism used to remove the Doppler shift. See below for more specific comments on this section.

1) On line 22 the authors mention that "...optical clocks promise advances in global navigation..." It is clear that optical clocks will support advances in geodesy and scientific measurements, but navigation on earth can already be performed at the centimeter level (depending on the exact configuration) and my understanding is that it is not limited by the clocks. If all GNSS clocks were replaced by perfect optical clocks today, it is not clear to me that one would see any improvement in GPS performance for instance. Do the authors have a reference that shows how optical clocks will change navigational performance?

2) Lines 114-116: From this sentence and later comments, it appears that the authors assume constant acceleration. Is that correct? If so, what would happen if the acceleration were not constant as would be the more general case?

3) Line 182: Please define "point-ahead" effects.

4) Line 207: It is common to use pseudo-random codes in TWSTT as a way of improving SNR. Why do the authors choose to use a uniform pulse rate instead?

5) Lines 207-208: The authors should state what the time offset is relative to? I believe it is to some "truth", which ultimately drops out and is not important, but is useful for understanding.

6) Lines 206-212: The authors should clarify the A/B nomenclature: is τ_A for instance the offset of A's clock? If so, doesn't the expression for t_A include B's clock errors since pulses arriving at A were sent from B? Just a concise definition is needed.

7) Lines 220-222: It appears that non-scripted I_{AX} is different from the scripted I_{AX} . I had to read this several times before I understood it. Apparently the scripted I_{AX} is the individual pulse shape and the non-scripted I_{AX} is the entire pulse chain. Maybe some more definition would be helpful or variable names that are more different from each other.

8) Lines 220-222: Does $t_{AX}[p_{AX}]$ mean the the AX time at index p_{AX} (the p_{AX} 'th interferogram)? The definition of terms is implied and probably could be inferred, but a concise definition would be helpful.

9) Line 222: The term "oracle timescale" should be defined. I assume you mean some absolute truth.

10) Paragraph starting at 231: This describes the use of the ambiguity function, typically used in pulsed radar, to determine the Doppler shift and propagation delay. This seems to be the key

advancement presented by the paper, yet it is only given a reference and not defined in the paper. This is probably adequate, but a clearer statement of what the ambiguity function is and how it is used in this context seems natural. Same comment for the Nelder-Mead search algorithm.

11) Line 238: I scanned reference (14) but didn't see the similar analysis referred to by the authors. The derivation of the times in line 239 was a bit opaque.

12) Line 244: What does it mean to have a non-integer ADC sample number? Does k simply index a particular sample, and if so how can it be non-integer, or does it mean something else?

13) In equation (5) are you simply extending eq. (4) to second order?

14) Line 261: repeat of an earlier comment: what if acceleration isn't constant?

15) The paper ends abruptly without a conclusion or summary?

We thank the reviewers for their constructive and kind comments on our manuscript, NCOMMS-18-37193. We have modified the manuscript in response to each comment. Below, we provide a point-by-point response.

As part of the response, we have addressed a general comment of reviewers 2 and 3 about expanding on some material from the Methods in the main text. To do this, we have broken the text after Eqn. 2 into three sub-sections headed “Velocity-Dependent Reciprocity Breakdown”, “Velocity-dependent Systematic Timing Shifts”, and “Synchronization feedback via an adaptive Kalman filter” with a more general new summary paragraph following Eqn. 2. In addition, we have expanded the “Conclusion” (originally mis-named “Discussion”) section to better summarize the key findings as well as provide more discussion on future applications.

Sincerely

Laura C. Sinclair, Jean-Daniel Deschenes, Nathan Newbury

Reviewer #1 (Remarks to the Author):

In the present manuscript the authors demonstrate a free space, optical two-way time and frequency transfer system that is robust against large change rates of the optical path length and achieves a precision compatible with optical atomic clocks.

The results presented in this work are certainly impressive and relevant to the time and frequency community. The claim that future optical clocks will require free-space optical time transfer may be bold but is not implausible. The free-space synchronisation and syntonisation technique developed by the authors and now demonstrated with a moving target is unique and world-leading.

The achievements presented in this paper are the cumulative result of several technological advancements, all of which have been (refs 13-15, 27, 28), or shortly will be (ref 32), published separately in greater detail. The challenges introduced by a moving clock site, which the authors choose to focus on, are not new, and the solution presented is conceptually straight-forward (see e.g. suggested reference below), albeit not necessarily easy to implement at this level of precision as the authors rightly point out.

As a whole, the achievement of femtosecond-level synchronisation via a flying quadcopter is a significant and noteworthy step towards practical free space, possibly satellite-aided, time and frequency transfer network for optical clocks.

The manuscript is generally well written and logical, though not always easy to understand. I have a number of specific technical comments – some of them may simply boil down to typing errors:

Line 51 – It is perhaps debatable whether the effects accounted for in this work should be called relativistic. Certainly, additional complications caused by clocks actually moving with respect to each other are not covered.

We agree with the reviewer that the additional complications caused by the clocks themselves moving are not covered here. We had tried to emphasize this point on lines 101-103. To respond to the reviewer's valid point, we have deleted that clause so that the final sentence reads "Here, we demonstrate an advanced comb-based O-TWTFT that synchronizes clocks to within femtoseconds despite motion".

Line 106 – The authors may want to cite Jeremy Warriner, John Hein, and Tom Celano and Richard Beckman, "Real-Time Two-Way Time Transfer To Aircraft", 38th Annual PTTI Meeting 2006 (<https://tycho.usno.navy.mil/ptti/2006papers/paper39.pdf>) or another suitable reference.

We thank the reviewer for pointing out this apropos reference. We have added it to the sentence on two-way rf transfer and also just prior to the non-reciprocity equation (2) since it derives the similar result for non-reciprocal transmission.

Line 107 – It is somewhat misleading to refer only to the second term as being caused by the finite speed of light, as the first term clearly also scales with 1/c.

We agree and have rewritten that section as follows: "The first term arises because the link distance is sampled at different times (asynchronously) by the pulses traveling each direction since they do not necessarily depart their transceiver at the same time. The second term exists even for synchronous sampling and arises from the motion of the retroreflector. It can be derived from geometric considerations of more formally via Lorentz transformations."

*Line 111 (also 46) - The number for the second term of the non-reciprocal correction, 1.3 ps, seems based on $L_A - L_B = 4$ km. Should this not be the *difference* of distances to the retroreflector, i.e. the presumably much shorter path segments between sites A and B, respectively, and the polarisation multiplexer? My understanding is that (most of) the free space path from each site to the mirror is shared.*

The reviewer is correct that it is the difference which is relevant. However, in some cases, we operated with two free-space optical terminals and a swept delay line located close to site B which would lead to the maximal possible error of 1.3 ps. We have modified this section to read:

"At a modest $V=30$ m/s, 4-km link, and an asynchronous sampling of $T_{AB} - T_{BA} + \Delta t_{AB} \approx 0.5$ ms, the first term in (2) yields a non-reciprocal time-of-flight of 50 ps. At the same velocity and for a reflector located adjacent to one clock site ($L_A - L_B \sim 4$ km), as is the case with the swept delay line, the second term in (2) yields a non-reciprocal time-of-flight of 1.3 ps. Note that for a symmetrically located retroreflector, as is approximately true for the quadcopter, this term vanishes."

Figure 1 – It would help with understanding, e.g. with the definition of events and timing calculations, if there was a schematic diagram of the full setup (sites A and B), not just site A.

Figure 1 is very information dense. For that reason, and because site B is almost identical to Site A, we had decided to display a schematic for only site A. We have modified the caption to highlight the nearly identical configuration of site B by adding the following sentence, “Remote site B is identical except that it contains only the remote comb B. In addition, the digital signal processor at site B applies feedback to remote comb B to synchronize the two sites.” We hope this change will be enough to address the reviewer’s point by providing more understanding for the reader. Note that it would be possible to also add a schematic of site B but the figure would have to grow in length significantly to fit all the information.

Line 153 – Was the same tracking technology as in ref 28 used in this work? I assume so, but it might be worth stating explicitly.

We have modified the sentence to read, “These data do show much longer fades due to the additional challenge of tracking the moving quadcopter, accomplished as in Ref. 30.”

Line 222 – I don’t know the term “oracle timescale”. I consulted two colleagues who didn’t recognise it either. Is it an ad-hoc, arbitrary time scale whose relation to well-defined timescale is determined retrospectively?

We thank the reviewers for pointing out the confusion that our use of “oracle timescale” caused. We have modified the text as follows: The time t is some common ad-hoc arbitrary timescale that is mathematically convenient but drops out of the final synchronization equation”. Elsewhere, we drop the term “oracle”.

Line 239 (equation 3) – Are the subscripted quantities on the left-hand side identical to those as in line 221 ($t_{AX}[p_{AX}]$ etc)?

The reviewer is correct. We made our notation uniform by replacing $t_{AX}[p_{AX}]$, $t_{BX}[p_{BX}]$, and $t_{XB}[p_{XB}]$ with t_{pAX} , t_{pBX} , and t_{pXB} .

Line 251 – Should the first time-of-flight on the right-hand side be labelled $T_{\{B \rightarrow A\}}$ rather than $T_{\{A \rightarrow B\}}$?

We thank the reviewer for checking our subscripts carefully. The subscripts in Eqn. 4 are correct as this equation represents the extrapolation of the A-to-B time-of-flight from one time set by $f_r^{-1}k_{pXB} - \tau_B$ to another time $f_r^{-1}k_{pBX} - \tau_A$. We have modified the text to read, “We first must extrapolate the time-of-flight to a common measurement time. To this end and to first order in V/c , we write:”

As a general remark, the authors do not refer to the accuracy of their technique, and they prefer to us the somewhat fuzzier term “precision” instead of “stability”. I wonder whether this could

be clarified throughout the text, to bring it more in line with accepted metrological terminology. Compare for example Lisdat, C. et al. A clock network for geodesy and fundamental science. Nat. Commun. 7:12443 doi: 10.1038/ncomms12443 (2016), which does use “precision” in places but also clearly analyses “stability” and “accuracy” separately.

We have attempted to clarify the language. The section header now reads “Analysis of time and frequency instability”. We have replaced the word “precision” with “uncertainty” in several places as uncertainty encompasses both the precision (instability) and accuracy. For example, “We include these two non-reciprocity corrections in (1) to <100 as uncertainty by using the available O-TWTFT data to calculate the speed to 20 $\mu\text{m/s}$ uncertainty at 1-second averaging time.” We avoided the word “accuracy” in our discussion of the site A-B clock offset since we are not connected to some absolute timescale and did not want to confuse the reader.

Reviewer #2 (Remarks to the Author):

The paper by Bergeron et al extends their free-space synchronization technology to cover a dynamic, small target with fs-level precision. The authors have pioneered the free-space time-frequency transfer and synchronization field in the last few years, and the current work overcomes the broken reciprocity due to a moving target. What is new in this paper is that synchronization algorithm is modified to suppress time-of-flight reciprocity breakdown, which is unique challenges for aerial dynamic targets. Data supports well all involved methodologies such as drone tracking, linear optical sampling, and suppressed timing asymmetry. Modified Allan instability for moving quadcopter after synchronization could achieve 10-18 instability. I recommend to publish this paper with minor revision, as the impact of the result could benefit overall metrology and applications of frequency combs involving aerial moving targets.

Some minor suggestions:

1. I recommend authors to plot timing offset of clock during signal fading in Fig. 3 (as shown in Fig. 2).

We have added the continuous measurement of the time offset of the clock and have also modified the figure caption accordingly. We have also modified the scaling of the y-axis of the bottom panel to encompass the clock walk-off during long duration fades. (To avoid issues of overplotting, the data are resampled to a 10 Hz rate.) The new Figure 3 is shown below. For clarity, it is now a two-column figure.

Figure 3: Synchronization results for a link to the flying quadcopter showing the pathlength (top panel), closing velocity (middle panel), and clocks' time offset (bottom panel) as measured by the out-of-loop verification channel. The clock's time offset is given for periods of active synchronization (grey dots) at the full ~ 2 kHz update rate and for all times (cyan dots) at a 10 Hz sampling rate. The latter clearly shows the walk-off of the clocks during longer duration fades. The walk-off of the clock offset can exceed ~ 100 femtoseconds for fades longer than 1 s. The standard deviation is 3.7 fs for active synchronization at the ~ 2 kHz update rate. (Also see Supplemental Video 1.)

2. As signal fades much frequently with quadcopter than with swept delay line, the role of Kalman filter based timing estimator is more important and more detailed discussion will be helpful.

We agree and have rewritten this section to add more details. The new section reads:

Synchronization feedback via an adaptive Kalman filter

The final synchronization algorithms are implemented in a digital signal processing platform to generate an estimate of Δt_{AB} in real time at a 2-kHz measurement rate. Under strong turbulence, signal fades block the exchange of comb pulses and the communication link, but these fades are usually of short duration, e.g. a few milliseconds. With motion, these signal fades can extend over longer durations because of the challenges of tracking the moving platform. Indeed, for the quadcopter data shown later, we suffer signal fades of many seconds. To synchronize despite these signal fades, we implement a Kalman-filter-based loop-filter as in Ref. 29. The Kalman filter provides optimal hold-over behavior during fades. It uses a two-element state vector, modeling respectively time error and frequency error. The state transition matrix is simply two integrators in order to accurately account for the random frequency-walk of the cavity-stabilized oscillators ($1/f^4$ phase noise). The measurement noise is modeled as white noise with a 5-femtosecond standard deviation at the 2-kHz update rate.

The Kalman filter's output is sent to a standard proportional-integrator controller which changes the Site B's frequency (via the offset between comb B and its local cavity-stabilized laser) to adjust the calculated time offset to zero. The combination of the measurement and clock noise levels sets the Kalman filter's effective bandwidth to 10-100 Hz, which limits the closed-loop bandwidth to the same value. During long fades, the clock time offset varies randomly according to the phase noise of the local cavity-stabilized laser. For our system phase noise, and in agreement with the Kalman filter model, the clock time offset can exceed 150 fs for fades longer than 1 s.

Reviewer #3 (Remarks to the Author):

This paper describes new work in synchronizing two optical frequency references over free space in the presence of an effectively large relative velocity, and associated Doppler shift. This group is famous for pioneering work in the area of free-space optical time and frequency transfer, but previous work has focused on comparing stationary sites. The extension to moving sites is a natural one and novel in the literature warranting publication.

The main result of the paper is overcoming the large Doppler shift between two sites and still maintaining 1e-18 precision in the comparison. Instead of moving the sites themselves, which would add unnecessary additional complexity, they use a moving transponder in the form of a quadcopter. Since the main idea is to introduce a real Doppler shift, which the moving transponder will do, the approach appears to be valid and is used over a wide range of velocities, demonstrating its robustness. Results are consistent over the range tested, providing confidence. The authors also use a swept delay line as an initial proof of principle and get similar results. The paper is only interested in the time/frequency transfer aspect of the problem. Having operating optical frequency references that are in motion is outside its scope.

The results appear to be valid and will be of interest to those working in this area. The work also appears to be the first of its kind, except for preparatory work by the same group.

The paper uses the Allan deviation to analyze the results and estimate final statistical uncertainty. These are the gold standards for this field and can't be argued with. The quadcopter data analysis appears to include the expected signal fades and drop outs caused by the intervening atmosphere and so addressing a more "real world" type of situation. As a result of the reasonable methodology and this statistical analysis, the conclusions appear to be valid.

I have the following comments that the authors should probably consider:

In general I found the paper easy to read except for the Method section. It seemed a little unusual to have it at the end, especially since it contains the key advancement of the paper, which is the mechanism used to remove the Doppler shift. See below for more specific comments on this section.

In response to this general comment and a similar one from Reviewer 1, we have added a brief discussion after equation (1) and subsection headers for the “velocity-dependent reciprocity breakdown”, “Velocity-dependent Systematic Timing shifts” and “Synchronization feedback via an adaptive Kalman filter”. As discussed below, we have added information or modified the wording in each of these to address comments of the reviewers.

1) On line 22 the authors mention that “...optical clocks promise advances in global navigation...” It is clear that optical clocks will support advances in geodesy and scientific measurements, but navigation on earth can already be performed at the centimeter level (depending on the exact configuration) and my understanding is that it is not limited by the clocks. If all GNSS clocks were replaced by perfect optical clocks today, it is not clear to me that one would see any improvement in GPS performance for instance. Do the authors have a reference that shows how optical clocks will change navigational performance?

We have changed this sentence to read, “... optical clocks promise advances in precision navigation, ...” to better reflect references 1-12. To address the reviewer’s comments, we agree that the current GNSS constellations are limited by several factors and a better clock would not solve everything. However, there are certainly efforts to place optical clocks in space and optical cross-links will assuredly follow. We have added a qualification to this future application of GNSS in the new conclusion.

2) Lines 114-116: From this sentence and later comments, it appears that the authors assume constant acceleration. Is that correct? If so, what would happen if the acceleration were not constant as would be the more general case?

Yes, in the analysis presented we assume constant acceleration. Our system returns the instantaneous velocity at a very quick update rate (~ 2 kHz). Further, we use centered derivatives to estimate the closing velocity at both sites at the correct time which greatly reduces the error contribution from acceleration. We have addressed the reviewers concerns as follows “Errors due to acceleration are suppressed to <0.1 fs at even our maximum experimental acceleration of 7g both by the fast update rate (~ 2 kHz) of the O-TWTFT system and the use of centered derivatives to determine the speed at the correct time.” As discussed below, we have added sentences to the Methods section on the changing acceleration: “A rough estimate of the time offset error due to a changing acceleration, i.e. jerk, is : $(1/6)(\dot{a}/c)(2\Delta f_r)^{-3}$. A 1 fs timing error is reached only at a jerk of $\dot{a}: 10^4$ g/s.”

3) Line 182: Please define “point-ahead” effects.

We have expanded the sentence to include a definition, “Additionally, ground-to-satellite links will suffer from non-reciprocity due to “point-ahead” effects, i.e. the separation of the up-link and down-link optical paths due to transverse motion, but theory and ground-based experiments indicate minimal impact²⁹⁻³¹.”

4) Line 207: *It is common to use pseudo-random codes in TWSTT as a way of improving SNR. Why do the authors choose to use a uniform pulse rate instead?*

The pulses are generated by frequency comb lasers which set a rigid repetition rate. Further, to measure these pulses, we use the technique of linear optical sampling which restricts our choice of waveform to purely periodic ones. Other techniques such as pseudo-random noise codes are currently unachievable with a comb-based, fs-jitter optical system at the 1-THz bandwidth used here. We agree it is a very interesting option to consider as frequency combs evolve further.

5 & 6) Lines 207-208: *The authors should state what the time offset is relative to? I believe it is to some “truth”, which ultimately drops out and is not important, but is useful for understanding.*

Lines 206-212: The authors should clarify the A/B nomenclature: is τ_A for instance the offset of A's clock? If so, doesn't the expression for t_A include B's clock errors since pulses arriving at A were sent from B? Just a concise definition is needed.

Yes, that is correct. We hope the new wording on the timescales (and removal of the word oracle time) discussed in response to reviewer 1 helps to address this issue. In addition, we have added the sentence “Note τ_A and τ_B are both here defined with respect to some common arbitrary ad-hoc timescale; in the end, we are only concerned with their difference.”

7) Lines 220-222: *It appears that non-scripted I_{AX} is different from the scripted I_{AX} . I had to read this several times before I understood it. Apparently the scripted I_{AX} is the individual pulse shape and the non-scripted I_{AX} is the entire pulse chain. Maybe some more definition would be helpful or variable names that are more different from each other.*

Yes, thank you. We have put a tilde over the symbol for the entire pulse chain.

8) Lines 220-222: *Does $t_{AX}[p_{AX}]$ mean the the AX time at index p_{AX} (the p_{AX} 'th interferogram)? The definition of terms is implied and probably could be inferred, but a concise definition would be helpful.*

In response to an earlier comment, $t_{AX}[p_{AX}]$ is now consistently denoted as t_{pAX} . We have added the definition “The times t_{pAX} , t_{pBX} , and t_{pXB} are the arrival time of the p_{AX}^{th} , p_{BX}^{th} , and p_{XB}^{th} interferograms.”

9) Line 222: *The term “oracle timescale” should be defined. I assume you mean some absolute truth.*

This was a concern of both reviewers #1 and #3 and we have addressed this under the comment from reviewer #1 above.

10) Paragraph starting at 231: *This describes the use of the ambiguity function, typically used in*

pulsed radar, to determine the Doppler shift and propagation delay. This seems to be the key advancement presented by the paper, yet it is only given a reference and not defined in the paper. This is probably adequate, but a clearer statement of what the ambiguity function is and how it is used in this context seems natural. Same comment for the Nelder-Mead search algorithm.

In response to the reviewer, we have modified the text (and included a reference to Nelder-Mead in the main body of the text). It now reads:

Velocity-dependent Systematic Timing shifts

The Doppler shifts of the received comb light are large (10^{-7} , or 20 MHz, at $V=30$ m/s) and changing as V is not constant. These Doppler shifts can couple with the system dispersion to cause distortions in the measured heterodyne signal between the incoming and local comb light. These distortions can lead to picosecond-level timing errors in the calculated timestamps. To avoid this, we calculate the cross-ambiguity function²⁷ between the measured heterodyne signal and a frequency-shifted template of the expected zero velocity waveform. We find its peak in real-time (<300 μ sec) to <100 as uncertainty by use of a Fourier transform algorithm and the Nelder-Mead search algorithm²⁸.

11) Line 238: I scanned reference (14) but didn't see the similar analysis referred to by the authors. The derivation of the times in line 239 was a bit opaque.

Thank you. We had referenced the wrong paper. This has been fixed.

12) Line 244: What does it mean to have a non-integer ADC sample number? Does k simply index a particular sample, and if so how can it be non-integer, or does it mean something else?

We have modified this section to explain as follows: “ k is the ADC sample number of the retrieved peak of the ambiguity function from the Nelder-Mead search, which is found with sub-sample precision. As a consequence, k is not restricted to an integer ADC sample number.”

13) In equation (5) are you simply extending eq. (4) to second order?

No. We apologize for the confusion as this was also commented on by Reviewer 1. We have rewritten this section to clarify the origin of equation (4) as discussed in the earlier response to reviewer 1.

14) Line 261: repeat of an earlier comment: what if acceleration isn't constant?

See comments under 2) above.

15) The paper ends abruptly without a conclusion or summary?

We thank the reviewer for pointing this out. We had accidentally mislabeled the “conclusion” section as “discussion”. We have fixed that error and expanded the conclusion section in the

revised manuscript.

REVIEWERS' COMMENTS:

Reviewer #1 (Remarks to the Author):

Thank you for responding to my numerous requests, as well the other reviewers'. I'm fully satisfied with the responses and the amendments made (where appropriate).

Reviewer #2 (Remarks to the Author):

The authors addressed the raised issues well. I recommend to publish the paper as it is.

Reviewer #3 (Remarks to the Author):

To both the authors and editors: I found the author's response to my (and the other reviewer's) comments clear, complete, and well thought out. I recommend publishing this article in the revised form without delay.

REVIEWERS' COMMENTS:

Reviewer #1 (Remarks to the Author):

Thank you for responding to my numerous requests, as well the other reviewers'. I'm fully satisfied with the responses and the amendments made (where appropriate).

Reviewer #2 (Remarks to the Author):

The authors addressed the raised issues well. I recommend to publish the paper as it is.

Reviewer #3 (Remarks to the Author):

To both the authors and editors: I found the author's response to my (and the other reviewer's) comments clear, complete, and well thought out. I recommend publishing this article in the revised form without delay.